# Caregiver Burden, Parenting Stress and Coping Strategies: The Experience of Parents of Children and Adolescents with Osteogenesis Imperfecta

**DOI:** 10.3390/healthcare12101018

**Published:** 2024-05-14

**Authors:** Alice Aratti, Laura Zampini

**Affiliations:** Department of Psychology, University of Milano-Bicocca, Piazza dell’Ateneo Nuovo 1, 20126 Milan, Italy; a.aratti@campus.unimib.it

**Keywords:** osteogenesis imperfecta, caregiver burden, parenting stress, coping strategies, perceived social support

## Abstract

Only a few studies, mainly qualitative thematic analyses of interviews, have dealt with the psychological experience of parents of children and adolescents with osteogenesis imperfecta (OI), a rare genetic syndrome involving skeletal fragility and increased exposure to bone fractures. The aim of the present study was to evaluate both negative (i.e., parental burden and parenting stress) and positive (i.e., coping strategies and perceived social support) experiences of parents related to their children’s disease and behaviour. The participants were 34 parents of children and adolescents with OI who completed a specifically developed online survey assessing their psychological experience with caregiving, their perception of the severity level of their children’s condition and any possible behavioural problems experienced by their children. Data analyses showed that 65% of the parents showed a clinical level of caregiver burden and nearly 30% a clinical level of parenting stress. Caregiver burden was related to the perceived severity level of the condition and the externalising problems shown by their children. Concerning the positive aspects of the parents’ experience, a high level of perceived social support was connected to a lower level of parenting stress; the same did not happen for caregiver burden. Coping strategies were connected to stress and burden; in particular, a higher level of stress corresponded to a higher level of avoidance, and a higher level of burden corresponded to a higher level of positive attitude.

## 1. Introduction

Osteogenesis imperfecta (OI) is a rare genetic syndrome with autosomal dominant transmission due to a mutation of the COL1A1 and COL2A2 genes with consequent skeletal fragility and greater exposure to bone fractures; for this reason, it is also called brittle bone disease. This mutation can be inherited from a parent with OI but can also occur spontaneously during conception [1]. OI affects approximately 1 in every 10,000 people [2], and in Italy, there are approximately 4000 people living with OI (www.telethon.it, accessed on 5 May 2024).

Every genetic disease is a potential stressor for dysfunctional outcomes in parenthood. Moreover, caring for a child with chronic illness involves the investment of additional psychological, emotional, social, physical and economic resources [3]. Therefore, for parents and caregivers of children with OI, living most of their lives in contact with such a complex medical condition represents a significant stressor [4]. In the case of OI, different studies have underlined that the aspects that make parenting a child with OI stressful could be many. Lazow and colleagues [5] found that the physical suffering and limited independence of children are the main risk factors for parents, who are highly likely to experience stress and a poor quality of life; Hill and colleagues [6] found that fractures, tiredness and mobility issues can limit parent–child interactions and activities and impact daily life and emotional well-being.

However, Hill and colleagues [7], in their systematic review of the impact of OI on family well-being, highlighted that families can face the challenges related to the disease and be resilient. The thirteen studies included in the review (one mixed-method study, six qualitative studies and six quantitative studies) involved about 500 participants, including 209 mothers and 65 fathers. Three fundamental themes emerged from the review: (1) the impact of OI on the psychosocial well-being of families, (2) the impact of OI on family life and (3) how OI is related to evolving roles and relationships. The review highlighted that fear of fractures and uncertainty about when the next one will occur are the main factors that impact family life and all family members. However, parents gradually accept the diagnosis and its consequences during child development with good adaptability. The children’s needs are incorporated into the daily routine, and the families generally demonstrate an excellent ability to maintain a positive perspective regardless of challenges.

Furthermore, the resilience of families with children with OI is one of the main findings of the research conducted by Hill and colleagues [6] on fifty-six participants, including eight parents of children with OI. The impact of OI on the daily life of patients and families was investigated through semi-structured qualitative interviews. The results highlighted that OI is not only a genetic condition but is also a factor affecting family functioning and parenting skills. However, despite the initial shock about the diagnosis, the parents explained how acceptance grew over time as they became used to dealing with the disease.

Social support is a further factor that increases the ability to cope with potential OI-related stressors since it might reduce the effect of anxiety, depression and post-traumatic stress symptoms. The descriptive and cross-sectional study conducted by Arabaci and colleagues [8] via face-to-face interviews with 46 parents pointed out that caregivers who obtain support from people close to them can develop greater hope for the future. In addition, access to the larger community of people living with OI through computer systems and the internet constitutes a significant resource of short- and long-term social support, as Castro and colleagues [9] demonstrated in their qualitative study on caregivers.

Although there is no research about coping strategies in parents of children and adolescents with OI, we know that parents of children with chronic illnesses can show two different patterns of coping strategies: adaptive strategies, such as acceptance, constructive coping and distraction, and maladaptive strategies, such as avoidance and denial [10,11]. However, it should also be noted that avoidance and denial could be adaptive strategies if the caregiver does not have the capacity to manage the stressor at the time [12,13]. A study carried out by Janusz and Walkiewicz [14] found three basic parental coping strategies related to children’s chronic illnesses: emotion-focused strategies, problem-focused strategies and avoidance-focused strategies. In general, adaptive strategies are associated with lower emotional distress and reduced burden, while maladaptive strategies are associated with greater distress and lower well-being.

### Aims of the Present Study

The main aim of the present study was to collect information about the psychological experience of mothers and fathers of children and adolescents with OI in order to analyse the impact of the disease (in terms of the severity of the condition and possible behavioural problems) on parental burden, parenting stress, parental coping strategies and perceived social support. The existing research in the literature about family well-being linked to children’s OI conditions is mainly of a qualitative type (e.g., thematic analyses of interviews) and focuses on a psychosocial description of the disease aspects [9,15,16,17,18,19,20,21,22]. The few quantitative studies about OI analyse specifically the negative aspects of living with the disease, such as stress, depressive symptoms or level of distress. Furthermore, they refer to Anglo-Saxon, American and Northern European contexts with no reference to other countries [8,17,23,24].

In the present study, we focused on quantitative data to outline a general picture of parents’ experience in the Italian context, with a focus on negative (i.e., parental burden and parenting stress) and positive aspects (i.e., coping strategies and social support) in parenting a child or an adolescent with OI. In particular, the study aimed to analyse the following: (1) the relationships between, on the one side, caregiver burden and parenting stress and, on the other side, the severity of the disease (as perceived by parents) and the possible behavioural and emotional problems shown by their children and (2) the relationships between parental negative experiences, in terms of caregiver burden and parenting stress, and coping strategies and perceived social support as possible resources.

## 2. Methods

### 2.1. Participants

The participants were 34 parents, i.e., 23 mothers (68%) and 11 fathers (32%), of children and adolescents diagnosed with OI. The parents were recruited from AS.IT.O.I (Associazione Italiana Osteogenesi Imperfetta), the Italian association for people with OI and their families. A public link from Qualtrics was made available via the OI association’s website and a newsletter to access the survey. Inclusion criteria were being a parent of a child or adolescent with OI ranging in age from 0 to 21 years and having a good knowledge of Italian, since the survey questions were written in Italian. Forty-one parents had access to the questionnaire, but four did not complete it in full.

The study was approved by the local commission for minimal-risk studies of the Department of Psychology of the University of Milano-Bicocca (RM-2021-454 on 14 October 2021). All parents signed an informed consent form before inclusion in the study.

### 2.2. Procedure

The participants were asked to complete a survey developed for the present study. The survey was created and distributed using Qualtrics (https://www.qualtrics.com). The questionnaire filling was anonymous and took about 30 min. The following two main topics were investigated:The features of children and adolescents with OI. In particular, the severity of the disease as perceived by their parents and the presence of any eventual behavioural or emotional problems;The parents’ psychological experiences, focusing on negative (caregiver burden and parenting stress) and positive aspects of life (coping strategies and perceived social support).

#### 2.2.1. Features of Children and Adolescents with OI

This topic was analysed through the following three dimensions:Personal data (e.g., age, sex) and anamnestic data concerning OI (e.g., type of OI). We asked the parents to focus on the last 12 months, and we requested them to indicate if their children were hospitalised (0 = no hospitalisation, 1 = one or more hospitalisations) if they had access to the emergency room (0 = no, 1 = one or more time) and if they had suffered fractures (0 = no fracture, 1 = one or more fractures). We chose to focus on the last 12 months because recent experiences may have a higher impact on the current situation.The severity of symptoms shown by children and adolescents diagnosed with OI, as perceived by their parents. To assess this dimension, the parents were asked to indicate if their children showed some problems in nine areas of development: (1) sight, (2) hearing, (3) autonomous breathing, (4) autonomous feeding, (5) self-care, (6) upper motor skills, (7) lower motor skills, (8) language and (9) cognitive abilities. For each of these areas, the parents were asked to rate on a five-point Likert scale if their children showed those problems or not as follows: (a) not at all (0 points), (b) a little (1 point), (c) enough (2 points), (d) much (3 points) and (e) very much (4 points). Then, a perceived severity score (Severity_score) (ranging from 0 to 36) was computed by adding the scores totalised in each area (e.g., a child with enough (2 points) difficulty in self-care and much (3 points) difficulty in language development would score 5 points in the severity score).Child/adolescent behavioural and emotional problems, as perceived by their parents, were assessed by The Child Behaviour Checklist (CBCL) [25,26], which analysed the children’s behaviour in terms of internalising and externalising problems. The questionnaire included two versions of the CBCL, one for the age range 1.5–5 years and the other for the age range 6–18 years, so the data were analysed according to the age of the children. In the present study, we considered two indices: one computed by adding the scores of the internalising scale (CBCL_internalising) and the other by adding the scores of the externalising scale (CBCL_externalising). The raw scores were then converted into T scores. T scores with a value equal to or greater than 70 indicated the presence of clinically significant behavioural or emotional problems.

#### 2.2.2. Parents’ Psychological Experiences

This topic was analysed through the following four dimensions:Parental burden, assessed by The Caregiver Burden Inventory (CBI) [27,28], which is composed of 24 items for evaluating the impact of the parental burden on different aspects of a caregiver’s life through 5 subscales: time-dependent burden (“I don’t have a minute’s break from my caregiving chores”) (CBI_Time dependent), developmental burden (“I wish I could escape from this situation”) (CBI_Developmental), physical burden (“I’m physically tired”) (CBI_Physical), social burden (“I’ve had problems with my marriage”) (CBI_Social) and emotional burden (“I feel angry about my reactions toward my care receiver”) (CBI_Emotional). In the present study, we also considered the index of total burden (CBI_Total). Scores between 0 and 24 indicated a low risk for burden, scores between 25 and 36 indicated a moderate risk and scores over 36 indicated a high risk for burden. The Italian version of the CBI has a high internal consistency (Cronbach’s alpha > 0.80).Parenting stress, assessed by The Parenting Stress Index short form (PSI) [29,30], which is composed of 36 items for measuring parenting stress through 3 subscales: parental distress, difficult child and parent–child dysfunctional interaction. In the present study, we considered the total stress index computed by adding the scores of the three subscales (PSI_Total). The total stress indexes were converted into percentiles, with a clinical range over the 85th percentile. The Italian version of the PSI has a good internal concurrent validity [30].Coping strategies of the parents, assessed by The Coping Orientation to Problems Experienced (COPE) [31,32], which is composed of 60 items for measuring coping strategies through 5 factors: social support (COPE_Social Support), positive attitude (COPE_Positive Attitude), problem solving (COPE_Problem Solving), avoidance strategies (COPE_Avoidance) and turning to religion (COPE_Religion). In the Italian version, the five scales have a high internal consistency (Cronbach’s alphas = 0.70–0.91).Perceived social support, assessed by The Multidimensional Scale of Perceived Social Support (MSPSS) [33,34], which is composed of 12 items for measuring environmental perceived support through 3 subscales: support from friends, support from family and support from a significant other (e.g., the partner). The questionnaire includes four items for each source of support (e.g., “My family really tries to help me”) on a 7-point rating scale ranging from 1 (very strongly disagree) to 7 (very strongly agree). In the present study, we considered the composite score of perceived social support obtained by averaging the three subscales (MSPSS_Total). Higher scores indicated higher levels of perceived social support. The Italian version of the MSPSS has a good internal concurrent validity [34].

### 2.3. Data Analyses

Statistical analyses were performed using IBM SPSS for Windows version 29. We used non-parametric tests since the sample size was small and the variables were not normally distributed. To investigate the first goal of the study, we used Spearman’s correlation and point biserial correlation to evaluate whether the children’s and adolescents’ problems (i.e., severity score and CBCL scores) or health conditions (i.e., hospitalisations, emergency room accesses and fractures) could influence caregiver burden and parenting stress. To investigate the second goal, we used the Spearman correlation to evaluate if perceived social support might be a protective factor against parental burden and parenting stress and whether coping strategies might be related to burden and stress.

## 3. Results

### 3.1. Description of Parents of Children and Adolescents with OI

The mean respondents’ age was 42 years (SD = 6.27; range = 29–55). All the participants were Italian, except for one mother from Switzerland. Italian was the mother tongue of all the participants. Concerning parental education, one parent (3%) attended elementary and junior school (8 years of education), five (14%) had a professional certificate of competence (11 years of education), sixteen (47%) had a high school diploma (13 years of education), seven (21%) had a master’s degree (18 years of education) and five (15%) had a postgraduate specialisation (19–23 years of education). Twenty-two parents out of thirty-four (65%) were working at the time of the study.

### 3.2. Description of Children and Adolescents with OI

Since in seven cases both the mother and the father of the same child participated in the study, the participants were the parents of 27 children and adolescents with OI ranging in age from 6 months to 19 years (M = 8.80; SD = 5.11). Fifteen (56%) were females.

Considering the type of OI, eleven children/adolescents had type I (41%) (i.e., the milder level), three had type IV-V (11%) (i.e., the moderate level) and eight had type III (30%) (i.e., the most severe level); the remaining five children/adolescents (18%) had type II-VII-VIII, which are new categories that are still little analysed. However, the type of genetic mutation is insufficient to determine the severity level of the disease in advance since the severity grading scale of OI relies on clinical and historical data, fracture frequency, bone densitometry and level of mobility of people with OI, as stated in a recent study confirming that the new OI nomenclature and the pre- and postnatal severity assessment emphasise the importance of phenotyping to diagnose, classify and assess the severity of OI [35]. The diagnosis was given in the prenatal period or at birth for 14 children/adolescents (52%) and during development for 13 children/adolescents (48%). The data regarding hospitalisations, emergency room admissions and fractures experienced during the last 12 months are reported in Figure 1.

Concerning how the parents perceived the difficulties and problems experienced by their children, the mean perceived severity score was 4.65 (SD = 3.79; range = 0–15) on a potential range of 0–36. The result was lower than expected. Analysing the nine areas of development, we found that a high number of parents reported difficulties in lower and upper motor skills and self-care; only a few parents reported difficulties in hearing, autonomous breathing, language and cognitive development (Figure 2). Therefore, the greatest difficulties were found in daily life’s practical and concrete activities rather than in intellectual abilities.

Concerning emotional and behavioural problems, as assessed by the CBCL, the parents reported a mean index of 64.17 (SD = 14.81; range = 39–95) for internalising problems. Five scores were within the borderline range (T = 65–70), and nine were within the clinical range (T > 70). Regarding externalising problems, the parents reported a mean index of 58.39 (SD = 12.52; range = 34–88). Six scores were within the clinical range (T > 70).

### 3.3. Descriptions of the Parents’ Psychological Experiences

Regarding the parents’ psychological experiences, we analysed the following negative and positive aspects of life: caregiver burden, parenting stress, coping strategies and perceived social support. The data are reported in Table 1. Concerning parental burden, the mean total score (M = 44.58) was above the cutoff that indicates a risk of caregiver burnout (>36), whereas concerning parenting stress, the mean total score (M = 58.75) was under the clinical range (>85). Twenty-two parents (65%) obtained a score indicating a high risk for caregiver burden, and ten parents (29%) achieved a clinically significant score for parenting stress.

Concerning coping strategies, only the mean score of positive attitude (M = 33.77) was over the norm score (30.9). Twelve parents (34%) scored above the norm score for social support, twenty-six (81%) for positive attitude, thirteen (37%) for problem-solving, six (18%) for avoidance strategies and fifteen (44%) for turning to religion. Concerning the perceived social support, the mean score was 5.42 (SD = 1.08) on a potential range of 0–7.

### 3.4. Relationship between Parental Difficulties and Children’s/Adolescents’ Problems and Health Conditions

To evaluate whether parental experiences and children’s/adolescents’ problems were connected, the CBI and PSI scores were correlated with the perceived severity level and the CBCL internalising and externalising scores. As reported in Table 2, total parenting stress (PSI_Total) was not significantly related to the perceived severity of symptoms or to the CBCL internalising and externalising problems, although a tendency towards significance emerged with the externalising scale. In contrast, caregiver burden (CBI_Total) was significantly related to the perceived severity level of symptoms (Severity_score) and to the externalising problems of the CBCL (CBCL_externalising). In particular, the time-dependent burden (CBI_Time dependent) was related to both the externalising problems of the CBCL and the severity of symptoms, whereas developmental burden (CBI_Developmental) was related only to the perceived severity of symptoms.

In addition, to evaluate if recent children’s/adolescents’ medical emergencies could influence their parents’ lives, we computed the correlations between the CBI and PSI scores and the occurrence of hospitalisations, emergency room admissions and fractures during the last 12 months. As reported in Table 3, only the total caregiver burden (CBI_Total) was significantly related to hospitalisations, since parents whose children had been hospitalised in the last 12 months showed higher levels of caregiver burden.

### 3.5. Relationship between Parental Difficulties and Positive Aspects of Their Experiences

To evaluate whether parental difficulties and positive aspects of their experiences were connected, the CBI and PSI scores were related to perceived social support and coping strategies. As reported in Table 4, overall social support (MSPSS_Total) was negatively significantly related to total parenting stress (PSI_Total). The negative correlation suggests that parents who experienced a greater level of perceived social support were found to be significantly less stressed.

Concerning coping strategies (see Table 4), avoidance strategies (COPE_Avoidance) were significantly related to parenting stress (PSI_Total). In contrast, a positive attitude (COPE_Positive Att) was unexpectedly significantly related to total caregiver burden (CBI_Total). The positive correlation suggests that parents who experienced a greater level of burden were found to use a positive attitude as a coping strategy (“I try to use this experience to grow as a person”, “I look for something positive in what is happening”).

## 4. Discussion

The present study was designed to collect caregiver ratings about the experience of parenting children and adolescents with OI in the Italian context. Both the negative and positive aspects of being a mother or father of children with OI were examined. The study focused on two goals: examining the relationships between parental difficulties (i.e., caregiver burden and parenting stress) and the children’s/adolescents’ features (i.e., severity of the disease and behavioural and emotional problems) and examining the relationships between parental negative experiences (i.e., caregiver burden and parenting stress) and possible resources (i.e., coping strategies and perceived social support).

A total of 29% of the parents showed a level over the clinical range in parenting stress, and 65% of them were at risk for caregiver burnout. These data confirm the difficulties experienced in parenting a child or an adolescent with a chronic illness, such as OI [4,36]. Living with such a complex medical condition represents a significant stressor that negatively affects the social lives of caregivers; in particular, parents caring for children with worse physical functioning are at higher risk for stress [7,8,16]. In addition, our results are in line with previous studies about caregiver burden: parents with children who experienced chronic illness reported a significant caregiver burden, especially because of the time spent on care to assist their children and the reduction in time and activities for themselves [37].

By analysing the effect generated by children’s and adolescents’ disease on parental burden and parenting stress, we found that the severity of symptoms was linked to parental burden, particularly time-dependent burden and developmental burden. Therefore, having a child who shows severe symptoms generates a restriction of one’s personal time (time-dependent burden) and a sense of failure regarding one’s hopes and expectations (developmental burden). Children’s/adolescents’ externalising problems were found to be linked to parental burden; this result is in line with previous studies about children and adolescents with neurotypical development and autism spectrum disorder that found a significant association between parenting stress and externalising behaviour problems in these populations [38,39,40,41]. Although not statistically significant, the relationship between parenting stress and externalising behaviours might be bi-directional, since parenting stress, deriving from a relationship with a difficult child, could induce anger and irritability in children and adolescents. In addition, parents with higher levels of stress might perceive their children as more problematic [42].

By analysing anamnestic data, we also found that parents whose children had been hospitalised in the last 12 months showed higher levels of caregiver burden. These data confirm previous qualitative studies on parents of individuals with OI that showed how a fear of fractures, hospitalisations and children’s behavioural problems generate stress and feelings of tiredness and worry connected to caregiver burden [9,21]. In particular, parents of children and adolescents with OI reported that they felt chronic exhaustion (physical burden), that their role in the family and towards other children had changed (social burden) and that they devoted most of their time to the children with OI (time-dependent burden); all these environmental factors analysed in qualitative research were connected to experiences of hospitalisations and fractures.

Concerning the positive aspects of parents’ experience, we found that a higher level of perceived social support was connected to a low level of parenting stress but not to a lower level of caregiver burden. It might be that the caregiver burden depends more on the children’s severity level and behavioural problems and might be less influenced by social support than stress. Having someone by their side could be a relief for parents, but it does not reduce their workload. The social support analysed in the present study concerned more emotional help (e.g., having someone to share positive and negative feelings or decisions) than material support (e.g., helping parents with daily life, home management, children’s therapies or medical tests).

Coping strategies appeared to be related to stress and burden; in particular, higher levels of avoidance corresponded to a higher level of stress, and higher levels of positive attitude corresponded to higher levels of burden. The relationship between stress and avoidance could be explained by considering that parents experiencing particularly stressful life situations, such as caregivers of children with disabilities, may detach themselves from problems by avoiding thinking about them or pretending they do not exist. When parents do not have enough resources to deal with the problem, denial can be an adaptive strategy [12,13]. These data confirmed previous studies in the literature suggesting that avoidant coping strategies are linked to the perception of bodily sensations considered unpleasant and anxiety-provoking [42] and that the use of avoidance, which is a maladaptive coping strategy, is associated with higher levels of stress in caregivers of children with autism spectrum disorder [43,44].

In contrast, the results of our study suggest that parents who experience higher levels of caregiver burden might implement a positive attitude, trying to use negative experiences to grow as a person and to accept the problem and live with it. This result is not consistent with previous studies on caregivers of stroke patients or individuals with dementia, which have found that the use of dysfunctional coping strategies contributed to increased feelings of caregiver burden and that adaptive strategies, such as a positive attitude, contributed to a decreased level of caregiver burden [45,46]. However, this result could be explained by considering the results of qualitative studies on the OI population; parents of children and adolescents with OI develop positive mindsets, optimistic views, coping strategies and resilience, so over time, the diagnosis and its consequences are accepted despite the problems and difficulties connected with the disease [9,17,18,20]. This situation of gradual acceptance is profoundly different from that of caregivers of individuals with dementia or other degenerative conditions.

### Limitations, Future Directions and Clinical Implications

The main limitation of this study is the small number of participants, which does not allow for a generalisation of the results to the highly variegated population of individuals with OI and limits the type of statistical analyses performed. Another limitation is that we detected the subjective perception of the parents and not the objective conditions of their children. Therefore, the parents’ answers might have been influenced by their current state of well-being. Moreover, since the survey was disseminated via the OI association’s website and newsletter, we have to consider that we analysed the experience of a particular subset of parents and that we had a non-response bias, since parents not following an associative life were probably under-represented.

In future studies, we will consider the relationships between subjective parental perception and the actual characteristics of children (for instance, correlating parental experience with medical reports) and will assess if participation in an associative life might relieve the feeling of isolation that often characterises rare genetic diseases [47] and, consequently, parenting stress and parental burden.

This study’s results shed light on the importance of supporting parents of children with OI to increase their well-being. A possible strategy would be implementing educational programs aimed at increasing functional coping strategies in these parents. Previous studies have demonstrated that specific parent training focused on the difficulties related to OI can lead to significant changes, including a reduction in feelings of anxiety and an increase in more functional and adaptive coping skills [7].

Moreover, the negative correlation between perceived social support and parenting stress underlines how it is important to analyse the context in which the families live. For instance, participating in a community of people with similar problems (e.g., associations of families) might have a positive impact on parent’s perceived social support. As shown by previous studies, having a connection with the larger community of people living with OI is an important source of social support [22].

## 5. Conclusions

Our results show that parents of children and adolescents with OI are at risk for caregiver burden and parenting stress. In particular, caregiver burden appears to be related to recent medical emergencies (i.e., having a child who has been hospitalised in the last 12 months), to the perceived severity of OI symptoms and to the externalising problems shown by their children. Parenting stress, but not caregiver burden, appears to be significantly mitigated by perceived social support; in addition, maladaptive coping strategies, such as avoidance, appear to be related to higher levels of parenting stress. These results suggest the importance of supporting parents of children and adolescents with OI to improve their quality of life.

## Figures and Tables

**Figure 1 healthcare-12-01018-f001:**
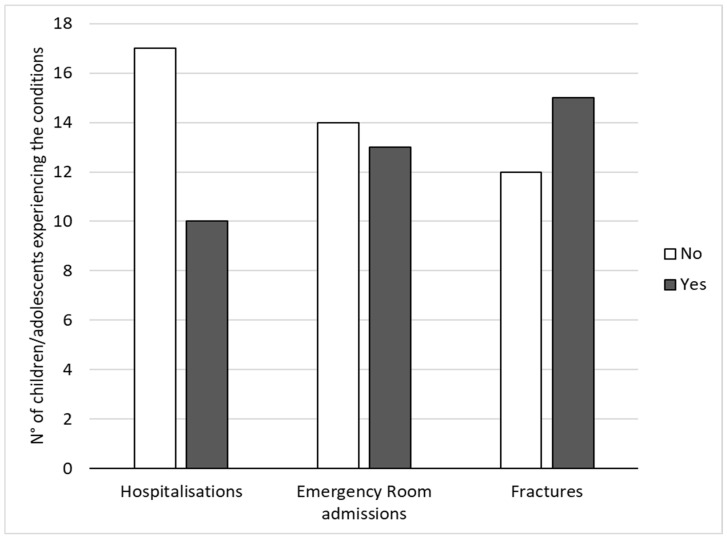
Number of children/adolescents experiencing hospitalisations, emergency room admissions and fractures during the last 12 months.

**Figure 2 healthcare-12-01018-f002:**
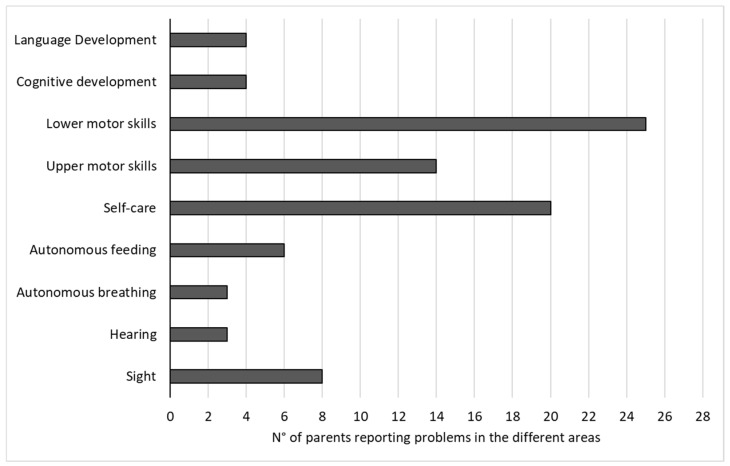
Number of parents reporting difficulties in their children in the developmental areas considered.

**Table 1 healthcare-12-01018-t001:** Descriptive statistics of caregiver burden, parenting stress, coping strategies and perceived social support in parents of children/adolescents with OI.

		M	SD	Range
Caregiver burden	CBI_Time dependent	14.12	5.16	5–25
	CBI_Developmental	8.18	3.97	5–23
	CBI_Physical	9.70	4.56	5–25
	CBI_Social	8.64	3.94	5–19
	CBI_Emotional	5.79	1.14	5–9
	CBI_Total	44.58	13.91	25–87
Parenting stress	PSI_Total	58.75	29.07	15–100
Coping strategies	COPE_Social Supp	25.67	6.74	13–37
	COPE_Positive Att	33.77	4.60	24–42
	COPE_Problem Solv	30.06	6.37	15–46
	COPE_Avoidance	20.47	3.81	16–33
	COPE_Religion	22.37	5.58	13–32
Perceived social support	MSPSS_Total	5.42	1.08	3–7

Note. CBI = Caregiver Burden Inventory; PSI = Parenting Stress Index; COPE = Coping Orientation to Problems Experienced; MSPSS = Multidimensional Scale of Perceived Social Support.

**Table 2 healthcare-12-01018-t002:** Relationship between caregiver burden and parenting stress and children’s/adolescents’ characteristics.

	Severity_Score		CBCL_Internalising		CBCL_Externalising	
	*Rho*	*p*	*Rho*	*p*	*Rho*	*p*
CBI_Time dependent	0.62	<0.001	0.24	0.203	0.39	0.039
CBI_Developmental	0.50	0.003	0.21	0.278	0.20	0.304
CBI_Physical	0.25	0.162	0.32	0.089	0.21	0.287
CBI_Social	0.22	0.215	0.18	0.340	0.24	0.228
CBI_Emotional	0.09	0.625	0.16	0.399	0.28	0.146
CBI_Total	0.54	0.001	0.30	0.111	0.39	0.043
PSI_Total	0.13	0.486	0.26	0.176	0.36	0.067

Note. CBI = Caregiver Burden Inventory; PSI = Parenting Stress Index; CBCL = Child Behaviour Checklist.

**Table 3 healthcare-12-01018-t003:** Relationship between caregiver burden and parenting stress with children’s and adolescents’ medical emergencies in the last 12 months.

	Hospitalisations		Emergency Room Admissions		Fractures	
	*rpb*	*p*	*rpb*	*p*	*rpb*	*p*
CBI_Total	0.43	0.013	0.07	0.688	0.31	0.075
PSI_Total	−0.01	0.955	−0.045	0.806	0.07	0.691

Note. CBI = Caregiver Burden Inventory; PSI = Parenting Stress Index.

**Table 4 healthcare-12-01018-t004:** Relationship between caregiver burden and parenting stress and coping strategies and perceived social support.

		CBI_Total		PSI_Total	
		*Rho*	*p*	*Rho*	*p*
Coping strategies	COPE_Social Supp	0.16	0.385	0.08	0.682
	COPE_Positive Att	0.38	0.037	0.30	0.114
	COPE_Problem Solv	0.32	0.073	0.23	0.214
	COPE_Avoidance	0.22	0.228	0.38	0.035
	COPE_Religion	−0.01	0.955	−0.15	0.429
Perceived social support	MSPSS_Total	−0.13	0.466	−0.58	<0.001

Note. CBI = Caregiver Burden Inventory; PSI = Parenting Stress Index; COPE = Coping Orientation to Problems Experienced; MSPSS = Multidimensional Scale of Perceived Social Support.

## Data Availability

The data that support the findings of this study are available from the corresponding author upon reasonable request.

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
