# Peer review of "Caregiver Burden, Parenting Stress and Coping Strategies: The Experience of Parents of Children and Adolescents with Osteogenesis Imperfecta"

_healthcare, 2024, doi:10.3390/healthcare12101018_

Round 1

Reviewer 1 Report

Comments and Suggestions for Authors

Comments and suggestions for Authors

I read with interest the manuscript entitled ‘Caregiver Burden, Parenting Stress and Coping Strategies: The Experience of Parents of Children and Adolescents with Osteogenesis Imperfecta’. The authors analyze the negative impact of their children's disease (Osteogenesis Imperfecta, OI) as well as the psychosocial factors that could counterbalance it (coping strategies and perceived social support).

I appreciate the authors' work and the systematized manner of presenting the research.

The Introduction includes information related to the impact of the children’s above-mentioned disease on the parents’ life and well-being. However, it should be evaluated if the specification of the number of participants from all the previous studies mentioned brings relevant information for the current study

This section may be improved by adding some aspects related to the coping strategies, given the fact that the authors took this aspect into consideration in their research. Even if coping strategies were not investigated in previous studies regarding parents with children with OI, studies that showed types of coping used by parents with other chronic conditions could be briefly mentioned.

It would also be useful to mention some data on the prevalence of OI, at least in the authors' country. This could be a starting point for some clarifications regarding the size of the group of participants studied (possibly in the Limitations section).

Methods’ section

Authors should check and correct the number of participants (37 parents in Abstract, line 14 and 34 parents in Methods section, line 95).  

Authors should provide more information regarding the recruitment of study participants (the method of sampling, the response rate (e.g. how many participants agreed to answer), inclusion/ exclusion criteria, etc.)

Lines 139-141 – authors should explain more clearly where the score 5 from the given example results from.

Lines 172-174 – Authors should write the full names of the first three scales/ factors of COPE inventory and then their abbreviations.

Results section

It should be checked if the numbers in the text correspond to those in the tables (e.g. line 230: M= 44.58 / table 1: M=44.21; line 231: parental stress M=58.75 / table 1: PSI=58.18; line 234: positive attitude (M = 33.77)/ table 2: COPE_Positive Att 33.72)

Legends should be added under the tables (it is customary to specify what the abbreviations mean).

Lines 257-259 – Authors should rephrase the sentence to be clearer.

Discussions section is well-organized.

Line 304 – maybe, for a better understanding, it should be specified that externalising problems refer to children.

Lines 334-336 - should be rephrased more clearly.

The Limitations, future directions and clinical implications section should be presented in a more organized manner (e.g. lines 370-371 refer to another limitation of the study, it should be presented after line 358). 

Authors should add a Conclusion section.

References list should be improved:

- out of the total of 41 cited references, only 18 are recent publications (within the last 5 years).

12.04.2024

Author Response

Reviewer 1

Thank you for your valuable suggestions. We have modified the paper, indicating the changed parts in blue in the text. Below, you will find the answers to your questions.

  1. I read with interest the manuscript entitled ‘Caregiver Burden, Parenting Stress and Coping Strategies: The Experience of Parents of Children and Adolescents with Osteogenesis Imperfecta’. The authors analyze the negative impact of their children's disease (Osteogenesis Imperfecta, OI) as well as the psychosocial factors that could counterbalance it (coping strategies and perceived social support).

I appreciate the authors' work and the systematized manner of presenting the research.

The Introduction includes information related to the impact of the children’s above-mentioned disease on the parents’ life and well-being. However, it should be evaluated if the specification of the number of participants from all the previous studies mentioned brings relevant information for the current study.

A1. We decided to report the number of participants in the previous studies since OI is a rare disease, and including a few participants could impact the results. The outcomes of studies with similar sample sizes could be better compared.

  1. This section may be improved by adding some aspects related to the coping strategies, given the fact that the authors took this aspect into consideration in their research. Even if coping strategies were not investigated in previous studies regarding parents with children with OI, studies that showed types of coping used by parents with other chronic conditions could be briefly mentioned.

A2. A paragraph on the coping strategies of parents of children with chronic illnesses has been added on page 2.

  1. It would also be useful to mention some data on the prevalence of OI, at least in the authors' country. This could be a starting point for some clarifications regarding the size of the group of participants studied (possibly in the Limitations section).

A3. Prevalence data has been added on page 1.

Methods’ section

  1. Authors should check and correct the number of participants (37 parents in Abstract, line 14 and 34 parents in Methods section, line 95).

A4. The number of participants in the abstract has been corrected. Thank you for noticing the mistake.

  1. Authors should provide more information regarding the recruitment of study participants (the method of sampling, the response rate (e.g. how many participants agreed to answer), inclusion/ exclusion criteria, etc.)

A5. The required information has been added on page 3.

  1. Lines 139-141 – authors should explain more clearly where the score 5 from the given example results from.

A6. This has been clarified on page 3.

  1. Lines 172-174 – Authors should write the full names of the first three scales/ factors of COPE inventory and then their abbreviations.

A7. Full names of the scales have been added.

Results section

  1. It should be checked if the numbers in the text correspond to those in the tables (e.g. line 230: M= 44.58 / table 1: M=44.21; line 231: parental stress M=58.75 / table 1: PSI=58.18; line 234: positive attitude (M = 33.77)/ table 2: COPE_Positive Att 33.72)

A8. Data in the tables have been checked and corrected.

  1. Legends should be added under the tables (it is customary to specify what the abbreviations mean).

A9. Legends have been added in notes under the tables.

  1. Lines 257-259 – Authors should rephrase the sentence to be clearer.

A10. The sentence has been rephrased to improve clarity.

Discussions section is well-organized.

  1. Line 304 – maybe, for a better understanding, it should be specified that externalising problems refer to children.

A11. It has been specified as requested.

  1. Lines 334-336 - should be rephrased more clearly.

A12. The sentence has been rephrased to improve clarity.

  1. The Limitations, future directions and clinical implications section should be presented in a more organized manner (e.g. lines 370-371 refer to another limitation of the study, it should be presented after line 358). 

A13. This section has been re-organised and also enriched following the comments of another reviewer.

  1. Authors should add a Conclusion section.

A14. A conclusion section has been added on page 11.

  1. References list should be improved:

- out of the total of 41 cited references, only 18 are recent publications (within the last 5 years).

A15. Recent references have been added to the paper.

Reviewer 2 Report

Comments and Suggestions for Authors

Dear authors, your work is valuable and complete in all their parts. The Background reflects the object of study, the references are appropriate, the study design is appropriate and the method adequate. The results reflect the discussions, as well as the conclusions are clear. I would have appreciated more attention to the statistical part, with the addition of a summary table or those key points and a final representation chart. Nevertheless, the work is excellent. 

Author Response

Thank you for your appreciation.

Reviewer 3 Report

Comments and Suggestions for Authors

The authors conducted an important study on the experiences of caregivers of children with Osteogenesis Imperfecta. The study contributes well to the body of research on caregiver burden. 

The study was generally well executed/reported. There are some areas that could be improved.

Methods

The methods section should contain a description of participant recruitment and data collection; the description of the sample (e.g., characteristics) belongs in the first paragraph of the results since you do not know these characteristics until after data have been collected and analyzed.

The procedural methods need attention. For example, describe how participants were recruited. How were participants invited?  How were they given the Qualtrics  survey? Were they screened?

Under measures, review the validity and reliability of the measures chosen.

Under data analysis, justify and reference the use of Spearman correlation. Consider regression (instead of Spearman) to explore the associations between several variables.  Include the sample size estimation.

Results

The first paragraph of the results should contain a description of the sample including mean age  of  caregivers and children,  ethnicity and so forth.   

The reporting of the statistically significant associations is unclear.

Limitations

The limitations section needs to be completed. The small sample size may also limit the type of analyses performed. Report other limitations such as selection bias, the reliance on  self-reports and memory, and so  forth.

Conclusion

There is no conclusion; please include a a conclusion. 

Comments on the Quality of English Language

Quality of writing

Generally the quality of writing was good. There is a problem, however, in some places with anthropomorphism. For example, it was written "The present study aims to collect information..." and studies cannot aim. 

Author Response

Thank you for your valuable suggestions. We have modified the paper, indicating the changed parts in blue in the text. Below, you will find the answers to your questions.

Reviewer 3

Comments and Suggestions for Authors

The authors conducted an important study on the experiences of caregivers of children with Osteogenesis Imperfecta. The study contributes well to the body of research on caregiver burden. 

The study was generally well executed/reported. There are some areas that could be improved.

  1. Methods

The methods section should contain a description of participant recruitment and data collection; the description of the sample (e.g., characteristics) belongs in the first paragraph of the results since you do not know these characteristics until after data have been collected and analyzed.

A1. The description of participants' characteristics has been moved to the Results section.

  1. The procedural methods need attention. For example, describe how participants were recruited. How were participants invited?  How were they given the Qualtrics survey? Were they screened?

A2. The procedural methods have been better described (see page 3).

  1. Under measures, review the validity and reliability of the measures chosen.

A3. Data on the validity and reliability of the measure chosen have been added.

  1. Under data analysis, justify and reference the use of Spearman correlation. Consider regression (instead of Spearman) to explore the associations between several variables.  Include the sample size estimation.

A4. We used non-parametric tests since the sample size was small and the variables were not normally distributed. For these reasons, regression is not suitable for the analyses. This has been explained in the data analysis section. This limitation has been added to the limitations section.

  1. Results

The first paragraph of the results should contain a description of the sample including mean age of caregivers and children,  ethnicity and so forth. 

A5. A paragraph with the sample description has been moved from methods to results.

  1. The reporting of the statistically significant associations is unclear.

A6. The paragraph has been rephrased to improve clarity (see page 7).

  1. Limitations

The limitations section needs to be completed. The small sample size may also limit the type of analyses performed. Report other limitations such as selection bias, the reliance on self-reports and memory, and so forth.

A7. The limitations section has been enriched.

  1. Conclusion

There is no conclusion; please include a a conclusion. 

A8. A conclusion has been added on page 11.

  1. Comments on the Quality of English Language

Quality of writing

Generally the quality of writing was good. There is a problem, however, in some places with anthropomorphism. For example, it was written "The present study aims to collect information..." and studies cannot aim. 

A9. The paper has been carefully checked.

Round 2

Reviewer 3 Report

Comments and Suggestions for Authors

The authors have made some important improvements.

A few issues remain.

Introduction (~ line 76) and discussion: it would benefit the readers to include a fulsome understanding of avoidance and denial as coping strategies that may be adaptive if the caregiver does not have the capacity to manage the stressor at the time (eg see Lazarus and Folkman).   

Under Methods, (~line 107) indicate where the public link was made available.   It seems there was no way to verify that only caregivers of children with OI completed the survey which should also be reported in limitations. 

The limitations are not fully reported (eg one notable missing limitation is non-response bias).  Please fully report all the limitations. 

Attention to quality and precision in writing still needed throughout the manuscript. For example, "Future studies will consider ..." Line 389 makes no sense since people consider things and people design studies. Line 307, it is stated "The  present study aimed to collect quantitative data..." and studies cannot aim as noted previously. Secondly, replace the word, quantitative" with a precise term such as  caregiver ratings  of their experiences. Similarly, in the abstract, "mainly qualitative," (line 9)  is written and should be replaced with a precise term such as interpretative research (or the design used to collect the qualitative data/experiences). 

Comments on the Quality of English Language

see above

Author Response

Reviewer 3

Comments and Suggestions for Authors

The authors have made some important improvements.

Thank you for your helpful comments. Detailed answers are given below.

A few issues remain.

  1. Introduction (~ line 76) and discussion: it would benefit the readers to include a fulsome understanding of avoidance and denial as coping strategies that may be adaptive if the caregiver does not have the capacity to manage the stressor at the time (eg see Lazarus and Folkman).   

A1. As suggested a reflection on avoidance as an adaptive strategy in parents of children with disabilities was added (see Lazarus & Folkman, 1984; Kelso et al., 2005).

  1. Under Methods, (~line 107) indicate where the public link was made available.   It seems there was no way to verify that only caregivers of children with OI completed the survey which should also be reported in limitations. 

A2. As reported in the manuscript, the link was made available on the OI website and sent to people who subscribed to the OI association's newsletter. The survey was explicitly addressed to parents of children and adolescents with OI, and one of the first questions was to indicate the diagnosis of their child.

  1. The limitations are not fully reported (eg one notable missing limitation is non-response bias).  Please fully report all the limitations. 

A3. The limitation section has been deepened. Thank you for your suggestion.

  1. Attention to quality and precision in writing still needed throughout the manuscript. For example, "Future studies will consider ..." Line 389 makes no sense since people consider things and people design studies. Line 307, it is stated "The  present study aimed to collect quantitative data..." and studies cannot aim as noted previously. Secondly, replace the word, quantitative" with a precise term such as  caregiver ratings  of their experiences. Similarly, in the abstract, "mainly qualitative," (line 9)  is written and should be replaced with a precise term such as interpretative research (or the design used to collect the qualitative data/experiences). 

A4. As suggested, we have corrected the “anthropomorphism” in the manuscript, even though this type of expression (e.g. “the study aims…”) is very common in scientific writing.

The word “quantitative” has been replaced by “caregiver ratings about the experience of parenting”.

The word “qualitative” has been replaced by “qualitative thematic analysis of interviews” (see the abstract and page 2).